# LoRaWAN Battery-Free Wireless Sensors Network Designed for Structural Health Monitoring in the Construction Domain

**DOI:** 10.3390/s19071510

**Published:** 2019-03-28

**Authors:** Gaël Loubet, Alexandru Takacs, Ethan Gardner, Andrea De Luca, Florin Udrea, Daniela Dragomirescu

**Affiliations:** 1LAAS-CNRS, Université de Toulouse, CNRS, INSA, UPS, 31400 Toulouse, France; alexandru.takacs@laas.fr (A.T.); daniela.dragomirescu@laas.fr (D.D.); 2Department of Engineering, University of Cambridge, Cambridge CB3 0FF, UK; elwg2@cam.ac.uk (E.G.); ad597@cam.ac.uk (A.D.L.); fu10000@cam.ac.uk (F.U.)

**Keywords:** wireless power transmission (WPT), wireless sensor network (WSN), simultaneous wireless information and power transfer (SWIPT), cyber-physical systems (CPS), structural health monitoring (SHM), Internet of Things (IoT), communicating material

## Abstract

This paper addresses the practical implementation of a wireless sensors network designed to actualize cyber-physical systems that are dedicated to structural health monitoring applications in the construction domain. This network consists of a mesh grid composed of LoRaWAN battery-free wireless sensing nodes that collect physical data and communicating nodes that interface the sensing nodes with the digital world through the Internet. Two prototypes of sensing nodes were manufactured and are powered wirelessly by using a far-field wireless power transmission technique and only one dedicated RF energy source operating in the ISM 868 MHz frequency band. These sensing nodes can simultaneously perform temperature and relative humidity measurements and can transmit the measured data wirelessly over long-range distances by using the LoRa technology and the LoRaWAN protocol. Experimental results for a simplified network confirm that the periodicity of the measurements and data transmission of the sensing nodes can be controlled by the dedicated RF source (embedded in or just controlled by the associated communicating node), by tuning the radiated power density of the RF waves, and without any modification of the software or the hardware implemented in the sensing nodes.

## 1. Introduction

By using Internet of Things (IoT) technologies that are now widely available, cyber-physical systems (CPS) can be implemented to respond to the needs of various applications. In the heart of these applications lies structural health monitoring (SHM) [1] for buildings, and civil and transportation infrastructures (e.g., railway and subway stations, bridges, highways, etc.) [2,3], which are part of the Smart City concepts and paradigms. In parallel, the building information modelling (BIM) concept [4] has provided new tools to more efficiently plan, design, construct and manage buildings and infrastructures during the different stages of their lifecycle. At the border between these concepts (SHM and BIM) the idea of ‘communicating material’ has arisen [5]. A ‘communicating material’ is intrinsically able to generate, process, store and exchange data from its environment to dedicated digital systems as a virtual model. A few works have already proposed implementations of communicating materials based on radio frequency identification (RFID) technology for various materials and applications. The main targeted application is the storage and access, at short ranges, of some disseminated or measured data in a material like wood [6], textile [7], or concrete [8].

The research project McBIM [9]—communicating material at the disposal of the building information modelling—aims to provide a practical application of the concept of communicating material in the construction domain through communicating reinforced precast concrete. Currently, reinforced concrete is the most common construction material thanks to its scalability, durability and cost [10]. The communicating concrete must be intrinsically able to generate, process, store and exchange data over several meters from its environment (i.e., the reinforced precast concrete element that is the monitored structure) to dedicated systems. These dedicated systems may include other structural elements made of communicating concrete (e.g., a floor or a wall) and a unique virtual model (a BIM), shared or transmitted between all the stakeholders. The system must be functional for the entire lifespan of the element (i.e., several decades) as part of a global structure and by itself.

As presented in Figure 1, the properties of precast concrete elements change regularly during the construction process of a structure, but often virtual/digital information is not shared or can be lost. Thus, a new owner cannot have a complete view of the history of their structural elements. By joining a unique virtual model (BIM) at each element, it becomes possible to conserve the entire history of an element and to create a virtual model of a structure and each of its components. The virtual model can also be updated at each modification (e.g., for each step in the lifecycle, for each owner change, etc.) and be partially stored in the element itself.

As presented in Figure 2, the risk of losing information is increased by the fact that many industries work together on a project and have similar tasks, but do not follow the same methods or process monitoring conventions.

Concerning the McBIM project, three main stages of the life cycle of a precast concrete element are targeted: (i) the manufacture and storage of each element; (ii) the construction of a structure by the combining of elements; and (iii) the exploitation of the structure, particularly through structural health monitoring. For instance, the monitoring of the curing process of a fluid screed or a precast element in terms of temperature and humidity could be a relevant use of communicating concrete during the manufacturing stage to help with the decision to continue the works (e.g., to unmould, to tile, etc.). The storage of environmental information of the element within itself and in the BIM allows the creation of a precise and real-time plan of a structure during the construction phase, whilst also continuously sharing information between all the stakeholders and track elements. The monitoring of the temperature at each side of a wall could be relevant during the exploitation phase to quantify the real thermal resistance of this wall and to certify some standards. Other uses, such as investigating maturity or applying preventative treatment through the detection of cracks, are also relevant in the construction domain. Thus, with a unique WSN several different applications can be met during the entire lifespan of the element according to the use of specific sensors.

Finally, literature that provides solutions for some specific applications with concrete is outlined. In [8] a solution based on RFID technology is proposed for monitoring moisture and temperature into concrete. The communication is short-range (centimetres) and the system only works when deployed and monitored by a user. This solution requires human interactions and therefore answers only a task during manufacturing and exploitation stages and cannot be fully automatized. A solution to monitor mechanical stress—through a load cell—and temperature of a concrete element is provided in [11]. This solution uses a battery which can be charged in the near-field thanks to an inductive wireless power transfer system and uses the M-BUS wireless protocol to communicate over a few metres. Again, human interactions are needed, here for the charging process. The tracking of precast concrete elements is provided in [12,13] by using embedded RFID tags in the concrete elements. In these cases, no measurement is performed, human interactions are needed for the short range reading, a smartphone connected to the Internet is required and only the construction phase of a structure is targeted. Several companies provide solutions to monitor concrete structures. For instance, [14] proposes a wireless temperature and humidity sensor with a battery based on SigFox (Labège, France) technology to be embedded into concrete and monitor its curing process for the two cases of in situ and precast casting. Some sensor nodes using batteries and middle range wireless communication are provided by [15] to characterize concrete in-situ. Finally, [16] provides an external and generic sensor to monitor concrete structure. All these solutions have a lifespan estimated at a few years, which is far less than the lifespan of a reinforced concrete structure.

In this paper, a smart wireless micro-node mesh network is proposed to design cyber-physical systems for SHM applications in the construction domain. This is the first step towards designing fully autonomous communicating concrete. The issues concerning installation and maintenance costs are considered, as well as the need of having a reliable system for decades. The wireless sensor network (WSN) presented in Figure 3 is composed of two kinds of nodes: sensing nodes (SNs) and communicating nodes (CNs).

The sensing nodes are designed to sense the physical world (i.e., the monitored structure) and communicate data collected from their environments to the communicating nodes over a distance ranging from several meters to kilometres. These nodes must be energy autonomous and reliable for the entire lifespan of the monitored structure because they cannot be accessed once deployed. Thus, to avoid any maintenance, and for increasing their lifetime, the SNs are designed to be battery-free and wirelessly powered by a RF source driven by (or even integrated into) the CNs through a far-field wireless power transmission (WPT) system. As communication and WPT work on the same frequency band, this architecture answers the paradigm of simultaneous wireless information and power transfer (SWIPT) [17,18].

The communicating nodes are designed to aggregate, process, store and exchange data generated by the SNs with the other CNs and with the virtual model through the Internet. As they are accessible, they do not need to be energy autonomous for the entire lifetime of the element. Moreover, the CNs are able to power the SNs by controlling the RF source dedicated to the WPT.

In this paper, an implementation of a LoRaWAN battery-free wireless sensing node is presented and characterized. It is powered via far-field WPT and measures temperature and relative humidity which are then communicated by using the LoRa technology and the LoRaWAN protocol to an ‘exploded’ communicating node (LoRa gateway and WPT system are currently separated). Its periodicity of measurement and communication is controlled by the WPT system through the tuning of the transmitted power. Thus, as the need of measurements cannot be fully specified during the design phase and can change during the different phases of the lifecycle of the element, the control of the periodicity through the WPT system allows the hardware and the software to remain unmodified. Moreover, some experiments were conducted with a network of two SN prototypes of sensing nodes.

Although our main objective is to provide a communicating reinforced precast concrete, the first experiments were carried out for generic in-the-air SHM applications and not yet in reinforced concrete. The current objective is to provide a proof-of-concept of a wirelessly powered and battery-free wireless sensing node designed for cyber-physical systems dedicated to structural health monitoring applications in the construction domain. This proof-of-concept must be powered wirelessly over several metres, must sense some relevant environmental parameters (in this case, temperature and relative humidity, but other types of sensors could be integrated according to the targeted application, such as mechanical stress sensors) and must exchange data wirelessly (in this case for long range, but some communication technologies are investigated according to the targeted application). In the near future, the prototypes will be embedded into a reinforced concrete beam to experiment with the targeted environment which is a more difficult medium for RF propagation.

Recently, several implementations of energy autonomous LoRaWAN sensing nodes dedicated to various applications, essentially for SHM, with various software or hardware defined periodicity, from seconds to hours, and with various kinds of sensors, such as temperature and humidity [19,20], mechanical stress [20,21], pH [19] or electrical current [22], have been presented. Some are battery-free [20,22,23], some harvest ambient energy (e.g., solar [19,21,23], thermal [19], mechanical [20], inductive [22], etc.) and others are based on backscattering [23,24]. Some solutions that combine multiple sources are proposed, too [19,21,23]. These projects are listed in Table 1. Nevertheless, and although the feasibility of WPT for powering a LoRaWAN node was demonstrated theoretically in [25], no complete implementation was provided until [26]. The advantage of WPT over ambient energy harvesting is that it is not dependent on the availability of the fluctuating ambient sources. Some estimates of available harvestable energy in buildings are shown in [27].

A complete specification of the implemented prototype of the sensing node will be the core of Section 2. Section 3 will highlight the obtained experimental results for a SN prototype and for a network of two prototypes of sensing nodes over the air. Discussions and future improvements will be discussed in Section 4.

## 2. Design and Implementation of the Prototype of Sensing Nodes

Due to its need for complete energy autonomy, the design of the sensing node is more challenging than the design of the communicating nodes. As previously described, the SNs must be battery-free and wirelessly powered, generate data by sensing their environment and transmitting these data wirelessly to the CNs.

### 2.1. Topology of the Sensing Nodes

Figure 4 shows the architecture chosen for the sensing node prototype. The SNs use a temperature and relative humidity sensor to sense their environment, which will be, in the end, reinforced precast concrete elements. The collected data are recorded by a microcontroler unit (MCU) in a LoRaWAN frame to be transmitted to the CNs through a wireless unidirectional radiofrequency communication based on LoRa technology. In order to help achieve energy autonomy, the SNs must be low power and as simple as possible. Therefore, they must carry out minimal data processing and should not store the collected data once transmitted. To provide the energy autonomy for decades without access, the SNs are battery-free and wirelessly powered through a far-field RF WPT system, which is controlled by the CNs. A rectenna is used in the SNs to harvest the dedicated generated RF power density and transform it into DC power which is supplied to a power management unit (PMU). This PMU has the role of efficiently managing the energy provided by the rectenna. It needs to store this energy in a supercapacitor and when there is enough energy to power all the active components (sensor, MCU and LoRa transceiver), deliver this energy to perform a measurement and a complete wireless transmission. As the SNs are inaccessible once deployed, no software or hardware modifications can be made. Nevertheless, by controlling the RF power density illuminating the SNs, the activation and periodicity of the measurement and transmission can be managed by the CNs.

### 2.2. Rectennas

To ensure the energy autonomy of the SNs, different strategies can be employed. As presented in Table 1, there are some published works on LoRaWAN sensing nodes based on energy harvesting techniques. The available energy sources and the best strategies to scavenge them in buildings and for SHM applications are highlighted in [27]. Unfortunately, these sources are too low, too fluctuating and sometimes unavailable for our targeted applications. For instance, light sources, air flows and thermal gradients are not available in concrete. The mechanical sources (e.g., vibrations) are fluctuating and hardly related to the uses of the structure (e.g., bridge, residential building, etc.). And the RF power densities measured in a laboratory in [27] are very low, specifically between 1.69 pW/cm^2^ in the 2.45 GHz ISM band and 57.37 nW/cm^2^ in the 900 MHz GSM band. Finally, the strategy of the RF wireless power transmission was chosen because of the complete control of the energy source and the possibility of managing the activation and periodicity of measurement and data transmission by tuning the transmitted power. There are two different approaches for RF WPT: the magnetic or near-field WPT as presented in [8,28,29] in the case of SHM of reinforced concrete structures applications, and the far-field RF WPT used in this project. The far-field solution can increase the useful range between the RF source and the sensing nodes. This maximum distance is related to the chosen frequency through the free space losses and to propagation medium through its dielectric properties. It should be noted that the capabilities of the far-field WPT systems are limited today mainly by regulations (i.e., the maximum effective isotropic radiated power (EIRP) that can be radiated in the ISM bands [30]) and not by the technology itself. For instance, the maximum EIRP for the ISM 868 MHz frequency band is 2 W (or +33 dBm).

In order to scavenge the generated RF power densities, specially designed rectennas (the acronym for “rectifying antenna”) are used. They are tuned to operate efficiently around the ISM 868 MHz band. The choice of this frequency is justified by a compromise between the free space losses and the size of the required antenna. The used rectennas, presented in Figure 5, are composed of:(i)a compact (11 cm × 6 cm) and broadband dipole antenna enclosed by a rectangular ring [31], manufactured on FR4 substrate (thickness: 0.8 mm, relative electric permittivity: 4.4 and loss tangent: 0.02) which captures and converts the generated electromagnetic energy field into a RF signal around the ISM 868 MHz band. Its simulated (HFSS—High Frequency Structure Simulator—from Ansys Inc. (Canonsburg, PA, USA)) maximal gain is +2.2 dBi at this frequency.(ii)a single Schottky diode (Broadcom Inc. (San Jose, CA, USA) HSMS 2850 mounted in series configuration) high-frequency RF rectifier (manufactured on Rogers Corporation (Chandler, AZ, USA) RT/Duroid 5870 substrate, thickness: 0.787 mm, relative electric permittivity: 2.3 and loss tangent: 0.0012) which converts the guided RF signal into DC power. For the first rectenna, the RF rectifier is connected with the antenna in an overlapping manner to have a planar two-dimensional (2D) rectenna. For the second rectenna, the RF rectifier is orthogonally connected with the antenna to have a three-dimensional (3D) rectenna.(iii)a metallic reflector (15 cm × 9 cm) positioned at 5 cm behind the rectennas (2D and 3D), which can increase the antenna gain (respectively from +2.2 dBi to +6.6 dBi) while having a three-dimensional structure comparable in terms of volume with the three-dimensional rectenna used in [32]. The manufactured rectennas are represented in Figure 6.

The use of the metallic reflector increases the rectenna performance by improving the directivity and the gain of the antenna in the opposite direction of the reflector. The adding of a metallic reflector increases the DC power harvested.

Undesirable harmonics are generated during the rectifying process, both at the input and output ports of the diode, which is a nonlinear component. An impedance matching circuit composed of a bent short-circuited stub and an inductor of 33 nH is inserted before the RF rectifier to be a band-pass filter and to avoid the re-radiation by the antenna of the generated harmonics. Thus, the RF power conversion efficiency is maximized. Again, a low-pass filter is inserted after the RF rectifier to provide a DC signal to the PMU.

The results depicted in Figure 7 demonstrate that the manufactured rectennas operate efficiently for low power densities of the incident RF waves. These generated power densities are higher than those available in buildings [27] and, consequently, a dedicated RF power source is required. Thus, DC powers greater than 50 µW can be harvested if the incident power density exceeds 0.42 µW/cm^2^ and 0.50 µW/cm^2^ respectively for the 2D and 3D rectenna. Moreover, these rectennas are relatively broadband, covering the ISM 868 MHz and 915 MHz bands, and a part of the GSM 900 MHz bands. Regarding Figure 7, the 2D rectenna is superior to the 3D rectenna with the use of the metallic reflector at 868 MHz. Nevertheless, both designs are within one order of magnitude in terms of DC power. More details concerning the rectennas used in this SN prototype are reported in [33].

### 2.3. Power Management Unit

To efficiently manage and store the DC power provided by the rectenna, a power management unit (PMU) is used. The Texas Instruments (Dallas, TX, USA) bq25504 [34] is chosen because of its minimum required input DC power (15 µW approximately). This PMU also contains a DC-to-DC convertor. Figure 8 shows the PMU operational behaviour. The PMU must efficiently charge the supercapacitor with the DC power provided by the rectenna and discharge the supercapacitor to power all the active components (the temperature and relative humidity sensor, the MCU and the LoRaWAN transceiver) when enough stored energy is available. To optimize its storage operation, a maximum power point tracking (MPPT) hardware function is implemented in the PMU. Regarding its operation, the PMU stores the available power until the voltage at the ports of the supercapacitor exceeds a selected threshold (V_max_ = 5.25V) and then supplies the active components until this same voltage goes down under another selected threshold (V_min_ = 2.30V). Moreover, both over- and under-voltage protections are used by the PMU. When the (under-) over-voltage threshold is attained the active components and the supercapacitor are disconnected in order to prevent a deep (dis)charge. The PMU integrates a cold-start procedure. Thus, if a minimal voltage of around 350 mV and a minimal DC power of 15 µW are provided to the PMU, it can be self-powered and begin the storage of the input power in the supercapacitor. Once enough energy is stored to reach almost 1.3 V voltage at the ports of the supercapacitor, the PMU switches on completely, optimizes its operation and more efficiently charges the supercapacitor. When a voltage higher than 1.3 V is available at the ports of the supercapacitor, the cold-start procedure is no longer necessary, even if there was no DC input power provided for a long period. The PMU intrinsic consumption—announced around 15 µW—is limiting regarding the minimum power that must be scavenged by the rectenna to allow the complete charge of the supercapacitor.

### 2.4. Supercapacitor

An AVX Corporation (Greenville, SC, USA) BZ01CA223ZSB supercapacitor [35] of 22 mF of capacitance (noted as C) is chosen in order to store enough energy to power all the active components during the required time for measurements (temperature and relative humidity) and the LoRaWAN transmission of a data frame. The supercapacitor exhibits a low loss current of 10 µA which must be take into consideration to estimate properly the required power that must be scavenged by the rectenna to power the PMU and to allow the complete use of the supercapacitor to store energy. The cycles of charges and discharges are managed by the PMU between V_max_ and V_min_ voltages. The energy stored (noted as E) by the supercapacitor can be estimated as:(1)$$E=\frac{C}{2}\cdot \left(V_{max}{}^{2}-V_{min}{}^{2}\right)$$

By using Equation (1), the energy stored by the 22 mF supercapacitor is estimated to 245 mJ in its voltage range (between V_max_ = 5.25 V and V_min_ = 2.30 V). Figure 9 shows the consumption of the SN prototype for a complete operation (measurement and wireless communication). Its consumption is estimated at 214 mJ. Thus, enough energy is stored by the PMU in the supercapacitor to power the sensor, the MCU and the LoRa transceiver during a complete operation.

### 2.5. Microcontroller Unit and LoRa Transceiver

To manage the temperature and relative humidity sensor, to capture the measured data in a LoRaWAN data frame and to control the LoRa transceiver, a microcontroller unit is used. The LoRa frame generated by the MCU is then transmitted by a LoRa transceiver. These processes are accomplished thanks to a Murata Manufacturing Co., Ltd. (Nagaokakyo-shi, Kyoto Prefecture, Japan) CMWX1ZZABZ-091 all-in-one LoRaWAN communication module [36] (composed of a STMicroelectronics (Schiphol, Amsterdam, Netherlands) STM32L072CZ [37] MCU based on an ARM Cortex M0+ and a Semtech (Camarillo, CA, USA) LoRa transceiver SX1276 [38]). A B-L072Z-LRWAN1 development board [39] by STMicroelectronics (Schiphol, Amsterdam, Netherlands) including the aforementioned hardware was chosen. As stated, the transceiver manages the LoRa physical layer protocol, whereas the MCU manages the sensor, the LoRaWAN protocol stack and the application software.

As presented in Figure 9, once powered, the MCU switches-on and initializes itself and all the other active components (the temperature and relative humidity sensor and the LoRa transceiver) (‘Init’ and ‘D’ stages). Then it manages the measurement from the sensor (providing two measurements encoded on two bytes) (‘M’ stage), processes the data and creates a LoRaWAN frame (17 bytes long, whose 13 bytes are induced by the LoRaWAN protocol) and drives the LoRa transceiver to send the LoRaWAN data frame (17 bytes long) (‘P and T’ stage). Once the complete data frame is sent, the system goes into a deep-sleep mode (‘D–S’ stage). All these operations are performed each time the SN prototype is supplied and the entire loop lasts nearly 2.13 s. The largest amount of DC power is required by the LoRa transceiver during the transmission of the LoRaWAN data frame. Nevertheless, for our applications, the measurements and the wireless communication must not be performed continuously (e.g., the temperature and the humidity should be measured and transmitted hourly or once a day/week, etc.). The transmission respects LoRaWAN class A standard (the node transmits data but cannot be interrogated), with a +2 dBm radio frequency output power at 868 MHz (ISM band), an activation by personalization (ABP) and without causing an acknowledgment from the network. This all aids to limit the number of exchanges and reduce the global power consumption. The bandwidth is 125 kHz and the spreading factor is 12, allowing a data-rate of 293 bps. It is probable that this configuration can be further optimized to consume less energy.

For the targeted application, the communication between SNs and CNs must be wireless and unidirectional (only uplink) in order to reduce the DC consumption by deleting the active reception process. The targeted distance between SN and CN is in the range of tens of metres (middle range) or kilometres (long range). LoRa technology and the LoRaWAN protocol are chosen for implementing our SN prototype because of: (i) the intrinsic long-range capabilities and (ii) the availability of a reliable infrastructure in the laboratory. Thus, a long-range (1.3 km) wireless communication between the SN prototype (located at LAAS-CNRS Toulouse, France, GPS coordinates: 43°33′45.3″ N 1°28′38.4″ E) and a LoRa gateway (installed on the INSA campus Toulouse, France, GPS coordinates: 43°34′14.8″ N 1°27′58.5″ E) can be performed. The data measured and transmitted by the SN are then routed by the TheThingsNetwork [40] network, up to servers, where these can be recovered and processed to obtain a complete cyber-physical system.

However, the choices of LoRa technology and the LoRaWAN protocol are not necessarily the best regarding the targeted application. Table 2 shows a short comparison between three technologies (LoRaWAN, Bluetooth Low Energy (BLE) and RuBee) identified as possible solutions to implement the wireless communication between the SNs and the CNs for different communication ranges (long and middle). It should be noted that the RuBee technology that is based on near-field communication, is not yet commercially available. It defines intrinsically an auxiliary channel for the WPT and is defined as insensitive to reinforced concrete and water. Conversely, BLE used the ISM 2.45 GHz band which is very sensitive to water and thus to concrete. Finally, LoRa is the most consuming solution among the three presented but with, in return, the longest reach. To estimate properly the power consumption of each standard, a hardware module must be selected and the duration of data communication (only uplink or bidirectional as function of the selected standard and/or application) must be considered.

### 2.6. Temperature and Relative Humidity Sensor

To measure the temperature and relative humidity, an Adafruit Industries (New York City, NY; USA) DHT22 [44] active sensor is used. Each measurement is coded on 2 bytes and has a resolution respectively of 0.1 °C and 0.1%. The measured average consumption of this sensor for its initialization and a unique measurement is 30 mW for a voltage varying from V_max_ to V_min_ or, in other words, nearly 64 µJ are needed to initialize and use this sensor for temperature and relative humidity measurements. This represents nearly 30% of the consumed energy and 26% of the stored and available energy. This sensor needs a ‘stabilized’ supply voltage between 3 V and 5 V during at least one second before performing its first measurement. Additionally, two seconds are needed between two consecutive measurements. The communication with the MCU is assured by a one-wire homemade protocol. Thus, this active sensor is not low-power, rather slow and is selected mainly by its availability and ease of use, with the objective to have a complete proof of concept. In addition, it is not dedicated to harsh environment and could not be embedded in concrete. More optimal sensing solutions are introduced and discussed in Section 4.3 as well as other kinds of sensor that could be relevant to integrate in the sensing node to monitor concrete elements.

### 2.7. Complete Operation

The supercapacitor is initially empty and the PMU is switched on by using a cold start-up procedure, which requires an input voltage higher than 350 mV and a DC power higher than 15 µW. When the cold-start is achieved and a DC power higher than 15 µW is available, the PMU charges efficiently the supercapacitor providing that its current loss is lower than the provided power. Once enough energy is stored, the active components are supplied. Thus, the supercapacitor is charged and discharged between two threshold voltages, as presented in Figure 8. Specifically, the software initializes the MCU, the sensor—with a delay of almost one second before being usable—and the LoRa transceiver. Then the SN prototype senses temperature and relative humidity and performs a LoRaWAN transmission, before being deactivated and shifting to a deep-sleep mode and then completely powering off. As presented in Figure 9, the complete operation lasts approximately 2.13 s (830 ms to initialize the system and make a temperature and relative humidity measurement and 1.30 s to transmit a LoRaWAN data frame). The system needs at least 214 mJ to work properly, where 64 mJ are for the sensor, as detailed earlier. The recovering, processing and storing of data on the servers has not been currently implemented. A visual check through a web-browser of the reception of the data frame is done on the TheThingsNetwork website.

A photograph of the two SN prototypes manufactured is shown in Figure 10. At this stage of the research the SN prototypes are either integrated or miniaturized but allow a proof-of-concept validation of the architecture highlighted in Figure 4. Moreover, the choices in terms of sensor and communication technology do not provide a real low-power solution. Nevertheless, having a functional complete SN prototype with the available sensor (not low power) and using LoRa technology and LoRaWAN protocol means that a more optimal system in terms of energy efficiency can be developed in the future for in the air applications. Moreover, its adaptation to be embedded in reinforced concrete must take into consideration new hard constraints especially regarding the attenuation induced by this particular medium. The following functionalities are currently implemented:
(i)the energy autonomy is performed by the WPT system (RF power source and rectennas), the supercapacitor and the PMU;(ii)the measurement is performed by a temperature and relative humidity sensor;(iii)the unidirectional wireless communication is warranted by the developed software implemented in the MCU and the LoRa transceiver; and(iv)the reconfigurability of the periodicity of measurement and data communication is possible by tuning the amount of transmitted power via the RF source.

## 3. Experimental Results

After having characterized each component individually (the twos rectennas, the PMU, the supercapacitor, the MCU and LoRa transceiver and the temperature and relative humidity sensor), a first complete SN prototype was entirely characterized, first by controlling the RF power at the input of the RF rectifier and then with a complete WPT system in an anechoic chamber. Some results obtained by using only one sensing node are presented in detail in [26]. However, the main results are recalled and completed in Section 3.1. Moreover, experiments were performed outdoors to check the transmission and reception of the LoRaWAN frames in more realistic environmental conditions. As proved in [26], an equivalence can be established between experiments performed inside the anechoic chamber (for which, as expected, no data transmission can be achieved between the SN and the CN—i.e., the LoRa gateway 1.3 km away—because of the presence of absorbents) and experiments performed outdoors (in a more convenient location—one balcony of our laboratory—for which the data transmission between the SN and the CN—i.e., the LoRa gateway 1.3 km away—can be easily achieved). Indeed, the DC profile consumption, the LoRaWAN data frame duration and the transmitted spectrum displayed on a spectrum analyser were identical for experiments performed in the two configurations: inside and outside the anechoic chamber. By using these equivalence criteria (observation of the DC profile consumption, the LoRaWAN data frame duration and the transmitted spectrum) we conclude that we can evaluate the quality of the data transmission by performing experiments only inside the anechoic chamber. No additional tests outside the anechoic chamber were performed at the date of submission. Some experiments with a minimalist network of two SN prototypes (each of them was equipped with a rectenna with metallic reflector) were conducted to validate their use in a network configuration and to certify that a unique RF power source is sufficient to power many sensing nodes in a defined area. Thus, a communicating node can control and recover the data frames from multiple sensing nodes. Then, a mesh network can be implemented between the CNs in order to store and exchange data in the area of an entire building and with a virtual model through the Internet to have a complete cyber-physical system.

In all experiments, a power synthesizer (Anritsu (Atsugi, Kanagawa Prefecture, Japan) MG3694B, operating at 868 MHz) is used as the RF source. To generate electromagnetic power densities that illuminate the rectenna and power the SN prototype in WPT experiments, a patch antenna with a maximal gain of +9.2 dBi is connected to the RF source. The losses due to the connection cables and connectors are estimated at –2.5 dB. The voltage measurements are performed using a LeCroy (Chestnut Ridge, NY, USA) WaveRunner 6100 oscilloscope. It must be noted that the oscilloscope measurements also induce current losses. Its input impedance is about 1 MΩ. Therefore, for a measured voltage of 5.25 V, 27.6 µW of power is lost (5.3 µW for 2.30 V).

As the transmission and WPT frequencies are the same (ISM 868 MHz band), linear polarized antennas in orthogonal polarizations are used for both functions. Thus, a practical solution for the SWIPT paradigm is proposed and tested.

### 3.1. Individual Sensing Node Prototype Characterization

#### 3.1.1. Experimental Results with Controlled RF Power at the Input of the RF Rectifier

For these experiments, the RF source was directly connected at the input ports of the RF rectifier equipped with a SMA connector. This RF rectifier was the one used next in the 2D rectenna.

As stated by the datasheet, the voltage required for completing the cold-start procedure and for switching on the PMU is around 350 mV. This voltage is reached when the RF power at input of the rectifier is at least –10 dBm. The cold-start duration is a function of the RF power provided to the RF rectifier. It is 3 h 33 min for a −8 dBm RF input and 17 min 51 s for a +0 dBm RF input. In the same way, the first charge duration (from an empty supercapacitor) lasts 5 h 16 min for a −8 dBm RF input and 29 min 4 s for a +0 dBm RF input. Figure 8 shows the typical voltage waveform observed at the ports of the supercapacitor for a complete process; from an empty supercapacitor to the first recharge.

Figure 11 presents the evolution of the measurement and transmission periodicity of the SN prototype and the output DC voltage of the RF rectifier as a function of the RF power at the input of the RF rectifier. As declared, the measurement and transmission periodicity of the SN prototype can be tuned by controlling the amount of the RF power provided to the RF rectifier and, thus, without hardware or software modifications. For this SN prototype and in this configuration, the periodicity can be imposed between 1 min 24 s for a +10 dBm RF input power and nearly 4 h for a −10 dBm RF input power (* and continuous line).

#### 3.1.2. Experimental Results with Controlled WPT System in an Anechoic Chamber

Experiments with a SN prototype powered by a WPT system positioned in an anechoic chamber were performed with the 2D rectenna with a metallic reflector. The experimental setup is presented in Figure 12. The anechoic chamber avoids any interference or multipath effects. There are 150 cm between the patch antenna of the RF source and the rectenna (harvesting the energy of the electromagnetic waves generated by the RF source) used to supply DC power to the battery-free SN prototype. Thus, the rectenna operates in the far-field region of the RF power source.

The RF power density illuminating the rectenna under test is computed as:(2)$$S=\frac{d^{2}}{3600\cdot \pi \cdot P_{T}\cdot G_{T}}$$
where P_T_ is the power generated by the RF power synthesizer, G_T_ the transmission patch antenna gain, and d the distance between the RF source and the rectenna. The RF power collected by the rectenna is computed as:(3)$$P_{RF}=G_{r}\cdot \frac{\lambda^{2}}{4\cdot \pi }\cdot S$$
where G_R_ is the gain of the antenna integrated in the rectenna and λ the wavelength. According to Equations (2) and (3), a power density of 0.5 µW/cm^2^ can be generated at 550 cm away from a 2 W or +33 dBm EIRP source.

Table 3 and Figure 11 (+ and dotted and dashed line) show the obtained experimental results. As the communication between the SN prototype located in the anechoic chamber and the LoRa gateway is impossible, the validation of a successful transmission is done by checking the SN prototype power consumption: if it goes into deep-sleep mode, the transmission is considered completed; and by the use of a spectrum analyser placed in the anechoic chamber in order to check the transmitted RF spectrum. As presented in [26], the communication from the SN prototype to the CN (LoRa gateway) can completely be checked (by recovering and decoding the transmitted frame from the Internet/TheThingNetworks) when the SN prototype is placed in a more compliant location (outside the anechoic chamber). Additionally, the measurement and transmission periodicity of the SN prototype can be tuned by controlling the power density illuminating the rectenna and thus without hardware or software modifications. For this complete SN prototype, the periodicity (between two consecutive sensing and data transmission) can be imposed between 5 min 41 s (power density of 4.16 µW/cm^2^) and 1 h 43 min (power density of 0.52 µW/cm^2^).

According to Figure 11, the results obtained during the two separate measurements—with a controlled RF power at the input of the RF rectifier and with a controlled power density illuminating the rectenna of the SN prototype—are consistent. The differences are in part due to the approximations in the estimation of the RF power at the input of RF rectifier integrated into the rectenna, which is estimated by taking into account the simulated gain of the antenna integrated into the rectenna and without taking into account a possible and unwanted mismatching and/or mutual coupling between the rectifier and the antenna.

### 3.2. Deployment of a Network of Two Prototypes

To deploy a mesh network as presented in Figure 3, a communicating node integrating an RF source, or an RF source controlled by the CN, must be used to wirelessly power several sensing nodes in its dedicated area (e.g., a part or an entire reinforced precast concrete element) and to recover the data frames sent by these SNs and maybe others in its neighbourhood. Currently, the RF source and the communicating node are separated. Nevertheless, the combination of the RF source and the LoRa gateway can be seen as an exploded CN.

The objectives of the experiments with a network of two SN prototypes are:(i)to check that a unique RF power source is sufficient to power many sensing nodes in a defined area;(ii)to have a first idea of the undesired interactions between the SN prototypes (i.e., coupling interferences, multipath effects, etc.); and(iii)to estimate the diversity in the performances induced by the intrinsic differences of the components (e.g., rectenna, PMU board, etc.) used to build the two prototypes. The SN prototype using the 2D rectenna with the metallic reflector is named Prototype no. 1 (SNP1) and the SN prototype using the 3D rectenna with the metallic reflector is named Prototype no. 2 (SNP2).

Some parameters/configurations were studied during these experiments:(i)both SN prototypes were tested alone in the same configuration to gain an idea of the intrinsic differences/diversity between SN prototypes;(ii)both rectennas of the SN prototypes were illuminated with the same power density and for two different values to have an idea of the diversity between SN prototypes;(iii)both rectennas of the SN prototypes were illuminated with the same power density to have an idea of the impact of the orientation of the rectenna according to the direction of the maximal gain of the antenna of the RF source; and(iv)both rectennas of the SN prototypes were illuminated with different power densities to have an idea of the impact of a SN prototype on the other.

The obtained results are gathered in Table 4 and plotted in Figure 11 (dotted lines). Figure 13 shows the experimental setup for both rectennas of the SN prototypes which were illuminated with the same power density of 3.16 µW/cm^2^. Figure 14 presents cold-start procedures and the first recharges of the two SN prototypes illuminated by the same power density (3.16 µW/cm^2^ in the anechoic chamber) and some charges and discharges. It can be observed that the two SN prototypes do not work in the same way although they are illuminated by the same power density. For a very similar DC voltage at the input of the PMU provided by the rectenna, prototype no. 1 is between three and four times slower than prototype no. 2. 

By analysing Table 3, Figure 11 and Figure 14, some conclusions can be made:(i)as expected, a unique RF source can efficiently wirelessly power several sensing nodes located in a dedicated area. The actual RF source (composed of the RF signal generator and the patch antenna) can generate a power density of 0.5 µW/cm^2^—which is enough to power the SN prototypes—at a maximum distance of 550 cm in the direction the maximal gain and at 390 cm for an angle of 30° for an EIRP of 2W or +33 dBm (maximum EIRP value authorized by regulation [30]), according to Equations (2) and (3).(ii)the acquired results are coherent with those obtained during the characterization of each prototype with some small discrepancies.(iii)there are some inconsistencies between the measurements performed with the two SN prototypes for which the only major difference is related to the rectennas used (2D rectenna with reflector plan for SNP1 and 3D rectenna with reflector plan for SNP2). This difference can be explained by the intrinsic diversity of the building blocks of our prototypes (especially the rectenna, the PMU and the supercapacitor), by the approximation in the calculation of the generated power densities and RF power at the input of the rectifiers and by the errors during the experiments and measurements.(iv)as expected, the orientation of the rectenna regarding the direction of the maximal gain of the RF source antenna has an influence by changing the ‘relative’ gain of the antenna integrated into the rectenna, with respect to the direction of the incoming electromagnetic waves generated by the RF source.(v)some interactions between the different SN prototypes were observed during the experiments. Prototype no. 2 was fixed and its rectifier output voltages, thus the periodicities of its measurement and transmission, are influenced by the location of prototype no. 1. Multipath effect and coupling can justify these observations.

For deployment in a reinforced concrete element, these effects will be accentuated by the nature of the concrete and by metal objects in the neighbourhood of the sensing nodes.

## 4. Discussions

The prototype of the sensing node -implemented and presented in this paper provides a proof of concept of a LoRaWAN battery-free wireless sensing node for SHM applications in the construction domain. The autonomy in terms of energy is guaranteed by the use of a wireless power transmission system, which can be tuned to control the periodicity of the measurement and communication. As it is neither integrated nor miniaturized, this SN prototype can be improved in order to be more complete, efficient, compact and relevant for the targeted applications. To date no experiments in reinforced concrete have been performed.

### 4.1. Improvements Achieved

From the draft introduced in [32] some remarkable improvements were achieved. The power management has been optimized by using a supercapacitor with a lower capacitance (22 mF for this work instead of 30 mF in [32], and this the lowest value from the comparable projects presented in Table 1 with [23]), lower loss current (10 µA for this work instead of 15 µA in [32]) and by well defining the voltage thresholds that control the charges and discharges of the supercapacitor (V_max_ and V_min_). Some refinements were performed regarding the rectenna. For a comparable volume (9 cm × 15 cm × 5 cm, or 675 cm^2^, for this work, by including the size of the metallic reflector, versus 6 cm × 11 cm × 7 cm, or 462 cm^2^, in [32]), the antenna of the rectennas used now has an higher maximal gain (+6.6 dBi in this work instead of +2.2 dBi in [32]) at the price of an increase of the overall volume of the prototype (+213 cm^2^ or 46%). By this increase of the maximal gain, the SN prototype can be powered with lower power densities illuminating the rectenna. Finally, the temperature and relative humidity sensor were added to have a complete LoRaWAN battery-free wireless sensing nodes powered by WPT. This addition increases the energy consumption by 64 mJ to complete all the operations (initialization, measurement, data frame creation and transmission). However, a lower capacitance is required, thus, the energy management implemented in this paper is more relevant and improved as compared with the solution presented in [32].

### 4.2. Future Work

In order to meet the needs of the communicating concrete some improvements must be made. First, a compact and integrated SN prototype must be provided. The use of a unique antenna—both for the WPT and the communication—could allow a high gain of compactness but would come with trade-offs that need to be considered (e.g., interferences, additional RF losses, etc.). The limitation in terms of volume is mostly due to the volume of the antenna and supercapacitor.

Concerning the energy management aspect, an optimization of the efficiency and a limitation of the losses can be accomplished by optimizing the rectenna. This aspect is very challenging because the input impedance of the PMU changes dynamically as a function of the input power and the energy stored in the supercapacitor. By increasing the rectenna gain, the SN prototype can be powered with lower power densities and, thus, the useful distance can be improved. An accurate estimation of the needed energy will allow refining the definition of the threshold voltages and, thus, limiting the supercapacitor capacitance. To be more efficient, an ultra-low power voltage regulator could be introduced at the PMU output in order to have a constant voltage to power the SN prototype. It could minimize the consumption of the LoRa transceiver because the higher the input voltage, the higher the consumption current. The consumption of the LoRa transceiver could be reduced by choosing a different configuration (e.g., bandwidth, spreading factor, etc.) according to the gateway capabilities.

Concerning the communication technology, the choice of LoRa technology and LoRaWAN protocol was dictated by the available infrastructure in the laboratory, the ease of development and the long-range capability. However, for a final SN prototype the choice of the communication technology will be function of some criteria as the targeted communication range between the SNs and CNs—between metres to kilometres—the power consumption, which must be low, the directionality, which must be unidirectional, the bit-rate or bandwidth and frequency. As presented in Table 3, Bluetooth Low Energy and RuBee could be relevant solutions for middle-range communication, but the frequency band used by BLE could be very constraining in concrete and RuBee is not currently commercially available.

Nevertheless, some projects have been ongoing during the last decade to provide sensing nodes powered by RF energy that use diverse communication technologies. These projects are listed in Table 5. EH stands for energy harvesting, FF for far-field and NF for near-field. Although they are broadly used and can be found in this short comparative study, RFID solutions are not discussed in this paper because of the limited communication range (typically tens of centimetres) in the ISM 868 MHz band. There is a large diversity of implementations of RFID technologies for SHM applications, especially through active tags [45] or tags which are intrinsically able to quantify a physical parameter [46]. Regarding projects of sensing nodes powered by RF energy, different approaches are proposed: the far-field energy harvesting in [47,48,49], the far-field wireless power transmission in [50,51,52] and the near-field or inductive wireless power transmission in [8] and that for a large variety of periodicity of the data transmission (from few seconds to tens of minutes). Moreover, various wireless technologies are used: proprietary or homemade in [47,48,49] or standardized as Bluetooth and BLE in [50,51,52] or M-BUS in [8]. Some prototypes are battery-free as [47,49,50,51]. Finally, although all projects sense temperature, various other kinds of sensor are presented as voltage [47,51], humidity [48,52], luminosity [49] or mechanical stress [8,52] for various applications especially around SHM.

Experiments must be conducted with the SN prototypes embedded in a reinforced concrete beam to validate the functionality of the WPT and wireless communication systems in a hard environment for the propagation of electromagnetic waves. The concrete beam, represented in Figure 15, has been manufactured in order to perform these experiments.

Concerning the choice of the sensor, a low-power, maybe passive, temperature and relative humidity sensor dedicated to harsh environments must be investigated. Moreover, temperature and relative humidity are not always the most important parameters and other sensors (e.g., stress, corrosion, etc.) should be integrated with regard to the targeted application. Mechanical stress sensors are, for instance, widely used in referenced projects [8,15,16,20,21,52]. Finally, a comparative study of available temperature and humidity sensors is discussed in Section 4.3.

### 4.3. Temperature and Humidity Sensor Choice

Due to the autonomous nature of this sensing node, it is essential to optimize every component that is embedded within. This section will carry out a comparison on various commercial ultra-low power temperature and humidity sensors all of which can be seen, with their properties, in Table 6. Included in this comparison is an in-house developed CMOS-MEMS thermodiode, which exhibits ultra-low power, ultra-wide range and long-term stability and linearity [53]. This will ultimately provide the knowledge to choose the best sensors for the application at hand.

It should be first noted that only some of the sensors from Texas Instruments (Dallas, TX, USA) [54,55] and from Robert Bosch GmbH (Gerlingen, Germany) [56] have the ability to measure temperature or humidity in isolation. This could be important when minimizing the power consumption or integrating multiple sensors. Due to the nature of the application, the accuracy is not the most important factor; however, the humidity accuracy for all the sensors is very similar and easily sufficient. This is the same for the temperature measurement accuracies, which are all under 0.5 °C. Here, the thermo-diode [53] possesses an advantage of measurement range, where it far exceeds any of the others. The most important factors, considering the autonomous objective, are the electrical parameters. The Sensirion (Staefa, Switzerland) sensors [57,58] claim to have the best average power consumption, which is defined as continuous operation with one measurement per second. This is arguably the most important factor when it comes to considering which sensor to opt for. Against this, the Texas Instruments (Dallas, TX, USA) sensors [54,55] have an average sleep current of 50 nA, which is an order of magnitude smaller than the rest. The measurement time is another parameter to consider, which is defined by the time of start-up and to take the first measurement. The Sensirion (Staefa, Switzerland) sensors [57,58] state that they can be power up and take a measurement within 1 ms, reducing the time from the current sensor by 800 times. The response time is unimportant for this application due to the long periods between measurements.

It is certain that implementing any of the above sensors can drastically reduce the power consumption and time that the system needs to be active; yielding great advantages. The Texas Instruments (Dallas, TX, USA) sensors [54,55] can individually measure humidity and temperature, which requires less power than measuring both. This gives the option of employing multiple sensors which could be beneficial to the reliability of the system for two reasons: (i) two measurements of temperature from separate devices provide a verification, and (ii) if one of the systems fails, there is still the other which can be back-up and provide pertinent information. The second benefit is especially important considering the desired longevity of the system. The thermo-diode [53] has been proven to last for hundreds of hours with DC current bias, which corresponds to decades of measurements. If the system uses the Texas Instruments (Dallas, TX, USA) HDC2010 [55] in conjunction with the thermodiode [53], the power consumption would be reduced by a factor of 100 whilst also boasting the benefits mentioned above. Considering all these factors, as well as the operation of the future system, will lead to the right sensor(s) chosen for optimum performance.

## 5. Conclusions

A wireless sensor network to actualize a cyber-physical system dedicated to structural health monitoring applications in the construction domain has been proposed in this paper. It consists of a wireless mesh network composed of battery-free LoRaWAN sensing and communicating nodes. A simplified implementation (for proof-of-concept purposes) consisting of two sensing nodes and one communicating node was experimentally tested over the air and presented in this paper. Temperature and relative humidity data are measured by two battery-free sensing nodes powered wirelessly by the same dedicated RF power source. The sensing nodes are switched on by using a cold-start procedure and start operating as long as they are illuminated with electromagnetic waves with a power density higher than 0.5 µW/cm^2^. The experimental results demonstrate that the data (temperature and relative humidity) collected by the sensing nodes can be transmitted wirelessly by using LoRa technology and LoRaWAN protocol to a communicating node, as a LoRA gateway located 1.3 km away from the sensing node. This communicating node interfaces our simplified sensor network with the Internet providing a bridge between the physical and the digital worlds. The autonomy in energy of the sensing nodes is obtained by using a wireless power transmission technique, which can be tuned to control the periodicity of measurement and data transmission without any software or hardware modifications of the sensing nodes.

Thus, we have proposed in this paper a complete implementation of a battery-free and wirelessly powered by far-field RF power transmission where sensing nodes can communicate over long distances by using LoRa technology and the LoRaWAN protocol. The sensing nodes measure temperature and relative humidity which can be controlled by their power transmission system. This prototype is currently working towards cyber-physical systems intended for structural health monitoring applications in the construction domain, with the ultimate goal of implementation within reinforced precast concrete.

## Figures and Tables

**Figure 1 sensors-19-01510-f001:**
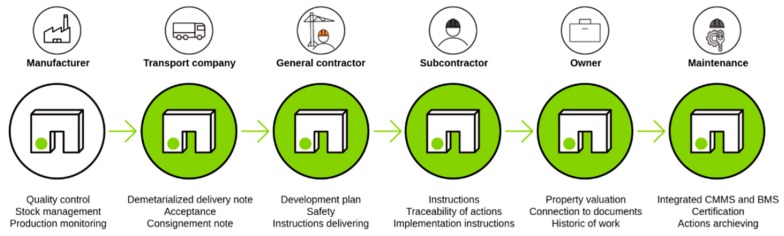
Changes of owner for a precast concrete element during the construction phase of a building.

**Figure 2 sensors-19-01510-f002:**
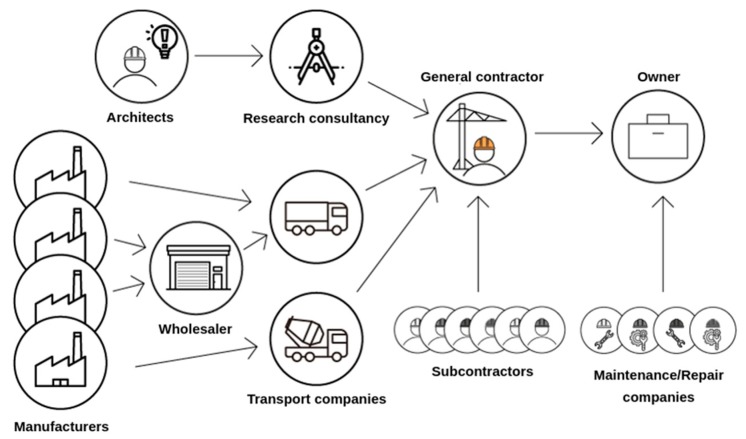
Representation of the interactions between the stakeholders working on the different tasks for a common building project.

**Figure 3 sensors-19-01510-f003:**
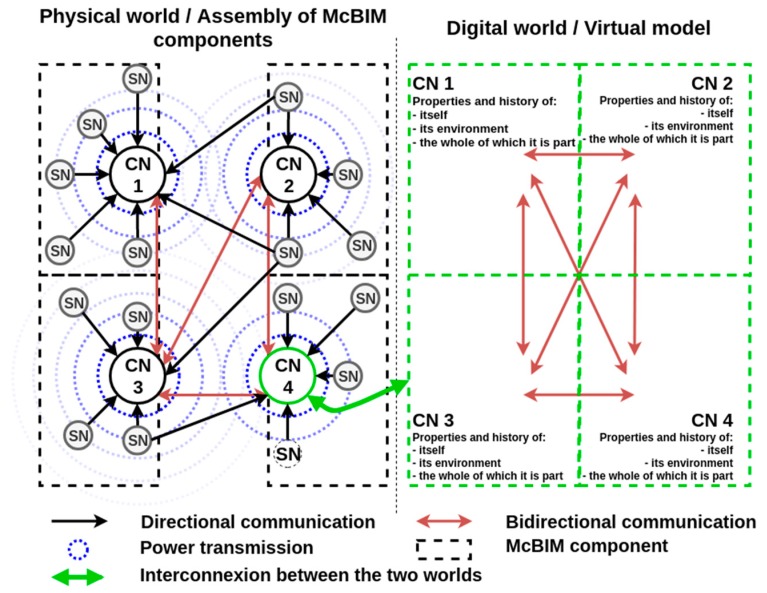
Schematic diagram of the architecture of the wireless smart-nodes mesh network.

**Figure 4 sensors-19-01510-f004:**
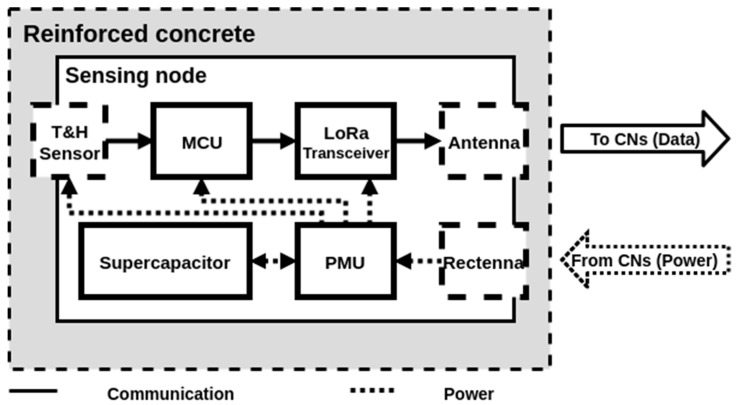
Schematic diagram of the architecture of the battery-free wireless sensing node.

**Figure 5 sensors-19-01510-f005:**
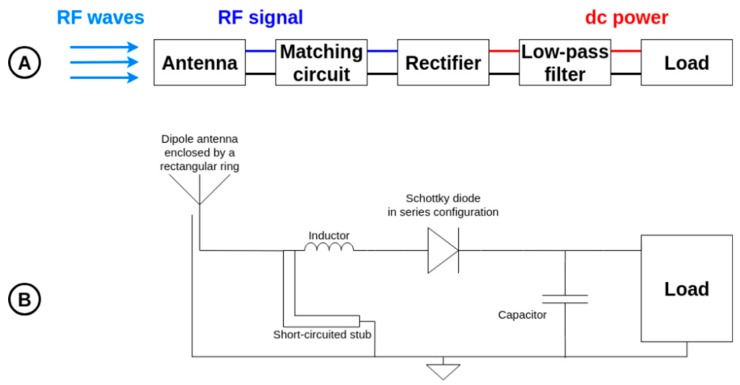
(**A**) Block diagram and (**B**) schematic of the manufactured rectenna.

**Figure 6 sensors-19-01510-f006:**
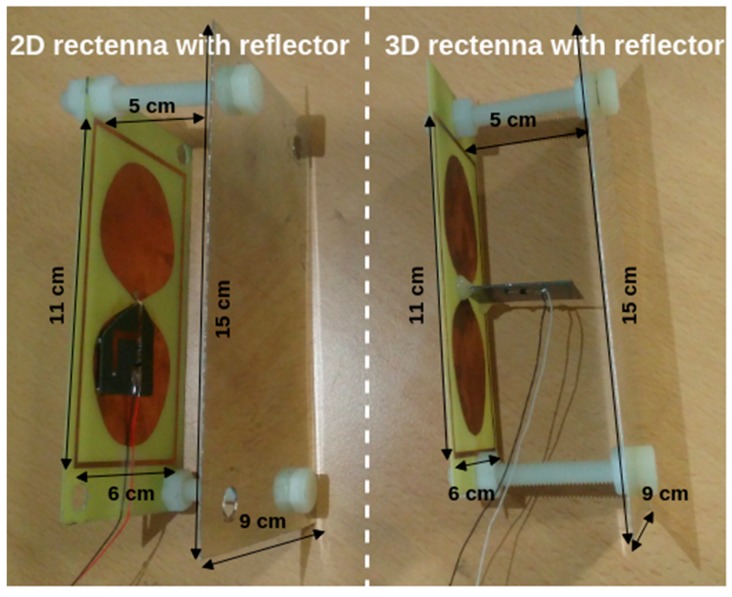
Photograph of the manufactured rectennas.

**Figure 7 sensors-19-01510-f007:**
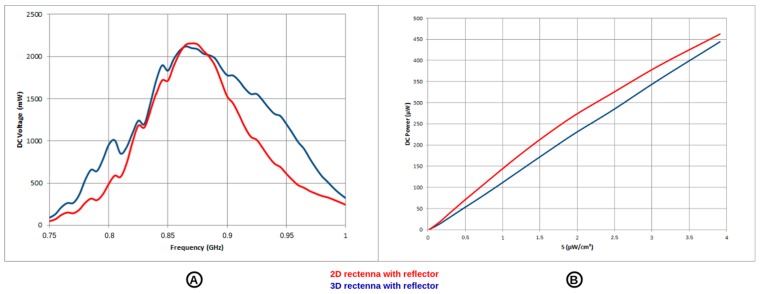
Measured rectenna performances: (**A**) DC voltage versus frequency (load: 10 kΩ, input power: +0 dBm) and (**B**) DC power versus illuminating power densities (frequency: 868 MHz, load: 10 kΩ); both the 2D-rectenna with reflector (red) and the 3D-rectenna with reflector (blue).

**Figure 8 sensors-19-01510-f008:**
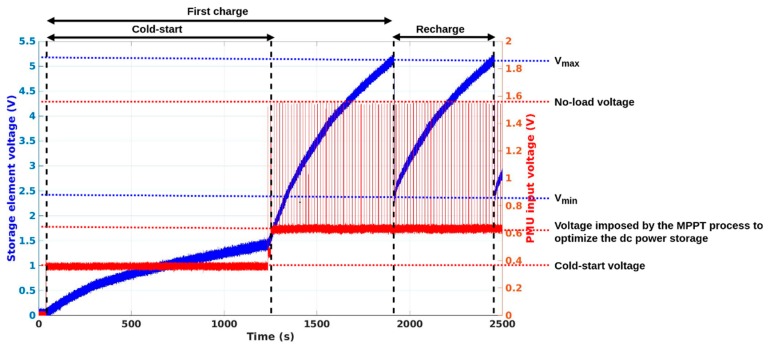
Voltage waveforms at the ports of the supercapacitor (blue) and rectenna output voltage (red), for a –8 dBm RF power at the input of the RF rectifier.

**Figure 9 sensors-19-01510-f009:**
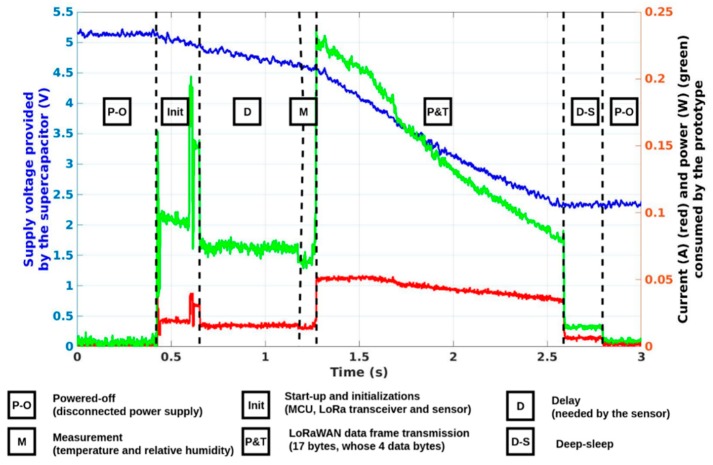
Supply voltage (blue), current consumption (red), and power consumption (green) of the SN prototype.

**Figure 10 sensors-19-01510-f010:**
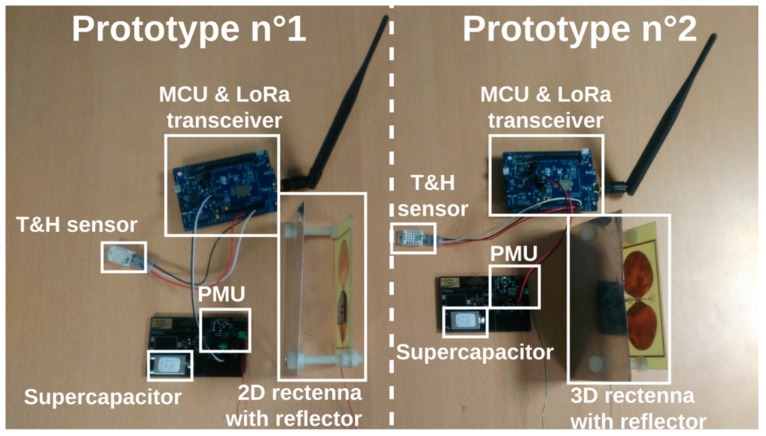
Photograph of the sensing node prototypes used for experiments.

**Figure 11 sensors-19-01510-f011:**
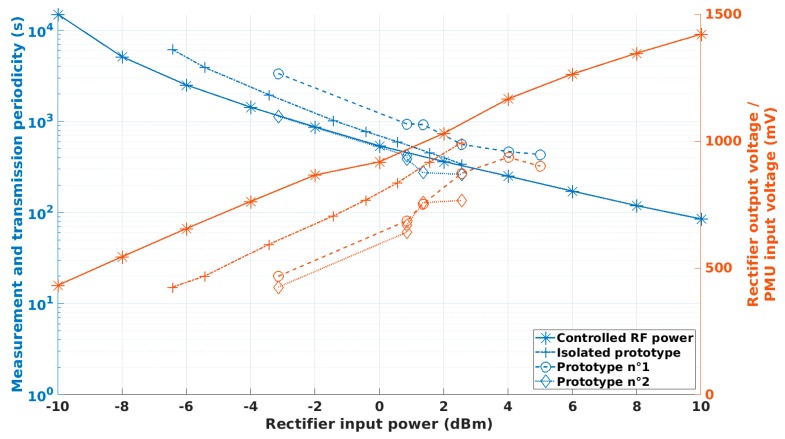
Measurement and transmission periodicity (blue) and rectifier output voltage (orange) as function of the applied RF input power for the sensing node prototype supplied by a controlled RF power at the input of the rectifier (*, continuous line), by a controlled power density illuminating the rectenna in an anechoic chamber (+, dotted and dashed line) and for the network of the two sensing node prototypes in the configurations presented in Table 3 (Prototype no. 1: o, dashed line; and Prototype no. 2: ◇, dotted line). In the case of a complete WPT implementation, the RF power at the input of the rectifier is estimated by taking into account the simulated gain of the antenna integrated into the rectenna, corrected with the angle.

**Figure 12 sensors-19-01510-f012:**
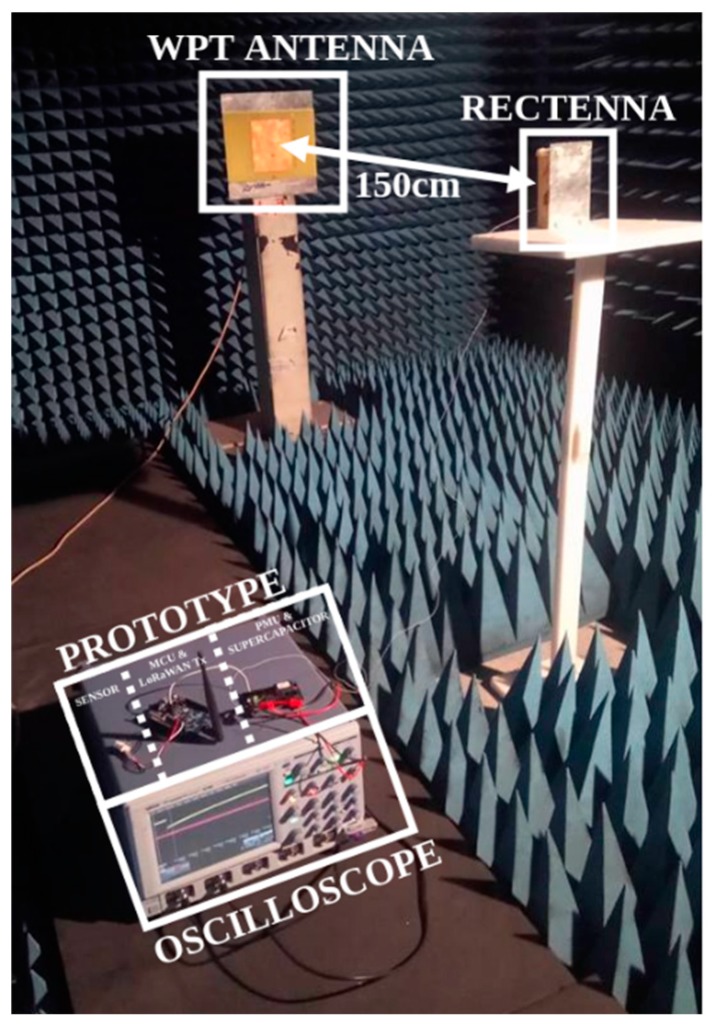
Photograph of the experimental setup for the sensing node prototype powered by using the WPT system in an anechoic chamber.

**Figure 13 sensors-19-01510-f013:**
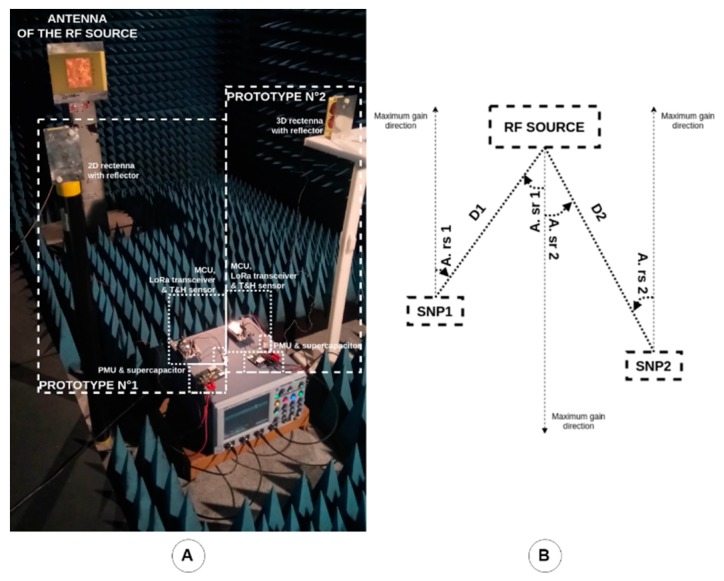
(**A**) Photograph and (**B**) diagram of the experimental setup for the network of two sensing node SN prototypes powered by using the WPT system in an anechoic chamber.

**Figure 14 sensors-19-01510-f014:**
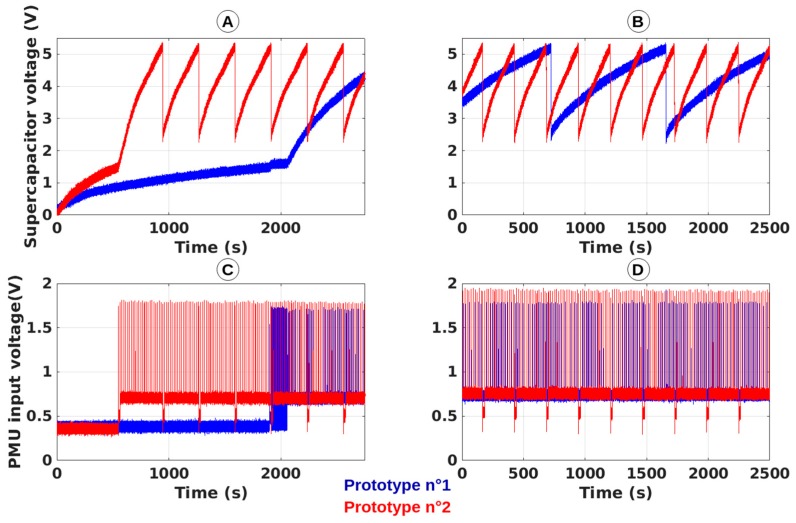
(**A**,**B**) Voltage waveforms at the ports of the supercapacitors of prototype no. 1 (blue) and prototype no. 2 (red) and (**C**,**D**) rectenna output voltages of prototype no. 1 (blue) and prototype no. 2 (red), for a 3.16 µW/cm^2^ power density illuminating the rectennas with (**A**,**C**) an angle of 20° and (**B**,**D**) an angle of 0°.

**Figure 15 sensors-19-01510-f015:**
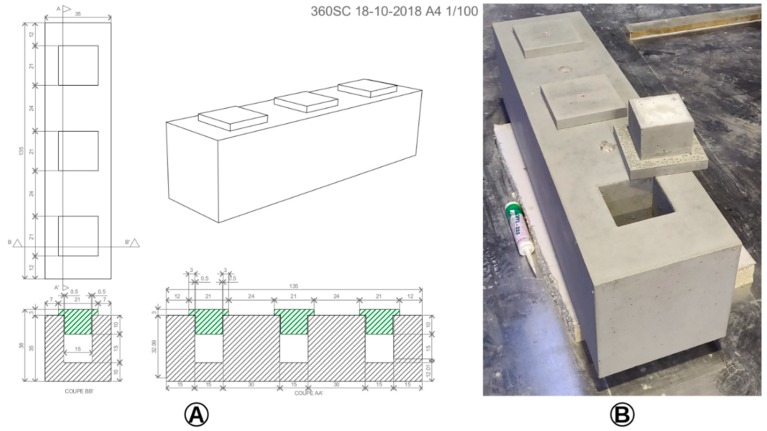
Reinforced concrete beam designed and manufactured to perform experiments in an environment targeted by the McBIM project: (**A**) design (dimensions in centimetres); (**B**) photograph of the manufactured reinforced concrete beam.

**Table 1 sensors-19-01510-t001:** Comparative study of implementations of LoRaWAN sensing nodes supplied by an energy-harvesting system.

[Reference] Year	Energy Source(s)	Energy Storage Capacitance	Sensor Type	Periodicity of Measurement and/or Transmission	Application(s)
[20] 2016	Mechanical (vibrations)	100 mF	Temperature Humidity (rain) Vehicles counter	3 h 30 min (estimation)	Bridges SHM
[24] 2017	Backscattering	N/A	N/A	N/A	SHM; Precision agriculture; Smart contact lens; Flexible epidermal patch sensor
[22] 2018	Electrical induction	22 mF	Electricity consumption/AC current	Seconds (few)	Smart grids
[21] 2018	Solar and battery	2 F and 1400 mAh	Crackmeter	N/A	SHM of construction materials
[19] 2018	Solar and thermal	20 Ah	Temperature Humidity pH	Hour	environment (water) monitoring
[23] 2018	Backscattering and solar	33 mF	N/A	20 min (estimation)	N/A
This work 2019	RF WPT	22 mF	Temperature Relative humidity	Controlled by the WPT system (minutes to hours, or more)	Communicating material SHM in harsh environments CPS

**Table 2 sensors-19-01510-t002:** Comparative study of three wireless communication technologies identified as possible solutions for the wireless communication between the SNs and the CNs.

Technology	Standard	Directionality	Frequency Band (ISM)	Range	Minimal Data Frame Size (byte)
LoRa	LoRaWAN [41]	Uplink (Downlink restricted)	433/868/915 MHz (far-field)	Long (km)	14
Bluetooth Low Energy (BLE)	IEEE 802.15.1 [42]	Bidirectional “Uplink-only” available	2.4 GHz (far-field)	Middle (tens of m)	11
RuBee	IEEE 1902.1 [43]	Bidirectional	131 kHz (near-field) (65 kHz for WPT)	Middle (tens of m)	5

**Table 3 sensors-19-01510-t003:** Experimental results obtained for the sensing node prototype powered by an RF WPT system in an anechoic chamber.

Power Density (µW/cm²)	Rectifier Input Power (dBm) [Estimation]	Rectifier Output Voltage (mV)	Measurements Periodicity
0.52	–6.43	424	~1 h 43 min
0.66	–5.43	468	~1 h 06 min
1.04	–3.43	593	~32 min 38 s
1.65	–1.43	705	~17 min 04 s
2.08	–0.43	767	~12 min 47 s
2.62	+0.56	835	~9 min 59 s
3.30	+1.56	917	~7 min 34 s
4.16	+2.56	994	~5 min 41 s

**Table 4 sensors-19-01510-t004:** Experimental results obtained for the network of two sensing node prototypes powered by an RF WPT system in an anechoic chamber.

Prototype no. 1							Prototype no. 2						
PD1^1^ *µW/cm^2^*	P1^2^ *dBm*	D1^3^ *cm*	A. sr 1^4^ *°*	A. rs 1^5^ *°*	V1^6^ *mV*	T1^7^	PD2^1^ *µW/cm^2^*	P2^2^ *dBm*	D2^3^ *cm*	A. sr 2^4^ *°*	A. rs 2^5^ *°*	V2^6^ *mV*	T2^7^
4.16	+2.56	150	0	0	873	9 min 19 s	/	/	/	/	/	/	/
/	/	/	/	/	/	/	4.16	+2.56	150	0	0	766	4 min 22 s
1.27	−3.14	150	20	20	467	55 min 48 s	1.27	−3.14	150	20	20	424	18 min 53 s
3.16	+0.86	150	20	20	686	15 min 36 s	3.16	+0.86	150	20	20	669	6 min 25 s
3.16	+1.36	150	20	0	752	15 min 25 s	3.16	+1.36	150	20	0	757	4 min 34 s
5.86	+4.01	110	20	20	936	7 min 46 s	3.16	+0.86	150	20	20	741	4 min 42 s
7.96	+5.01	75	30	30	901	7 min 13 s	3.16	+0.86	150	20	20	641	7 min 01 s

^1^ Power density illuminating the rectenna of the SN prototype (estimated by using Equation (2)); ^2^ Rectifier input power (estimated by using Equation (2)); ^3^ Distance between the RF source and the rectenna of the SN prototype; ^4^ Angle between the direction of the maximal gain of the RF source and the location of the rectenna of the SN prototype; ^5^ Angle between the direction of the maximal gain of the rectenna of the SN prototype and the location of the RF source; ^6^ Rectifier output voltage; ^7^ Measurements periodicity.

**Table 5 sensors-19-01510-t005:** Comparative study of sensing nodes powered by RF energy harvesting or RF WPT.

[Reference] Year	Energy Source(s) (Distance/Frequencies)	Energy Storage Capacitance	Communication Technology (Frequency/Range)	Sensor Type	Periodicity of Transmissions
[47] 2010	FF RF EH from TV broadcast (hundreds of meters or kilometres/VHF or UHF)	100 µF	SimpliciTI (2.4 GHz/middle)	Temperature Voltage	5 s
[48] 2012	FF RF EH from ambient sources (ISM band) (meters/2.45 GHz)	(Battery)	N/A (433 MHz/m)	Temperature and humidity	45 s
[49] 2013	FF RF EH from TV broadcast or cellular base station (hundreds of meters or kilometres/VHF and UHF)	160 µF	N/A (2.45 GHz/m)	Temperature and luminosity	[1 min; 4 min 30 s]
[50] 2016	FF RF WPT (metres/868 MHz)	1.8 mF	BLE (2.45 GHz/m)	Temperature	30 min
[8] 2017	Inductive WPT (centimetres/100 kHz)	(Battery)	M-BUS (169 MHz/hundreds of meters)	Stress and temperature	When powered
[51] 2018	FF RF WPT (tens of centimetres/868 MHz)	1 mF	Bluetooth (2.45 GHz/m)	Temperature and voltage	[5 s; 20 s]
[52] 2018	FF RF WPT (meters/868 MHz)	N/A	BLE (2.45 GHz/m)	Displacement Temperature and humidity	0.5 s
This work 2019	Far field RF WPT from dedicated RF source (meters)	22 mF	LoRa/LoRaWAN (868 MHz/km)	Temperature and relative humidity	Controlled by the WPT system (minutes to hours, or more)

**Table 6 sensors-19-01510-t006:** A comparison of various commercial humidity and temperature sensors from information available on datasheets. Average power consumption is defined as continuous operation with one measurement per second. Measurement time is the time to start-up and record an initial measurement. * Speculated value.

Reference		Adafruit DHT22 [44]	Texas Instruments [HDC2080] [54]	Texas Instruments [HDC2010] [55]	Bosch [BME280] [56]	Sensirion [SHTW2] [57]	Sensirion [SHTC3] [58]	ThermoDiode [53]
Functionality	Temperature	Yes	Yes	Yes	Yes	Yes	Yes	Yes
	Humidity	Yes	Yes	Yes	Yes	Yes	Yes	No
	Measure in Isolation	No	Yes	Yes	Yes	No	No	N/A
Humidity Characteristics	Average Accuracy (± %RH)	3	2	2	3	3	2	N/A
	Specified Range (%RH)	0 to 100	0 to 100	0 to 100	0 to 100	0 to 100	0 to 100	N/A
Temperature Characteristics	Average Accuracy (± °C)	0.5	0.2	0.2	0.5	0.3	0.2	0.1
	Specified Range (°C)	–40 to +80	–40 to +85	–40 to +85	–40 to +85	–30 to +100	–40 to +125	–200 to +600
General Characteristics	Response Time (s)	2	8	8	1	<5 to 30	<5 to 30	<1 *
	Average Power Consumption (µW)	2800	1	1	3.3	8.6	16	<1 *
	Sleep Current (µA)	/	0.05	0.05	0.1	0.7	0.3	/
	Measurement Time (ms)	800	-	-	2	1	1	<10 *

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
