# Peer review of "LoRaWAN Battery-Free Wireless Sensors Network Designed for Structural Health Monitoring in the Construction Domain"

_sensors, 2019, doi:10.3390/s19071510_

Round 1

Reviewer 1 Report

Please refer to Guideline for Authors to unify the language.

Is there any using restriction in the study?

In this study, the main contribution and application of experiment results in concrete should be expressed clearly.

This study mentioned the short-term observations for transmission, therefore, It should get strongly convincing and evidence in long-term observation.

In this study, please describe in view of the relevance references to support it originality.

Author Response

Dear Reviewer,

The authors would like to thank you for your given time and your helpful comments on this manuscript. We hope to have carefully addressed all the comments and revised the manuscript accordingly. The point by point replies to the comments and the main changes in the revised paper are summarized below.

Sincerely,

The Authors.

==========================================

Reviewer 1: Comments and Suggestions for Authors

Please refer to Guideline for Authors to unify the language.

è          We unified the language as recommended in the Guideline for Authors.

Is there any using restriction in the study?

è          To date, because it is the beginning of the project, the study is carried on for generic in-the-air applications (and not in reinforced concrete). The main objective is to provide a proof-of-concept of a wirelessly powered and battery-free wireless sensing node designed for cyber-physical systems dedicated to structural health monitoring applications, especially in construction domain. This proof-of-concept must be powered wirelessly over several meters, must sense some relevant environmental parameters (in a first time, temperature and relative humidity, but next other type of sensors could be integrated according to the targeted application) and must exchange data wirelessly (in a first time, for long range but other communication technologies are investigated according to the targeted application and reinforced concrete constraints related to RF propagation).

               Our final objective is to embed prototypes into reinforced concrete, but for ease of development, we decided to begin with in-the-air experiments and to introduce our work as a generic solution for CPS in SHM applications. Next step is to experiment in the targeted environment that says in a reinforced concrete beam, which will be more constraining for RF propagation. To date (March 2019) we have no conclusion about a real use in reinforced concrete but works/research (very time consuming) are under run.

               Moreover, the powering distance is related to the minimum power needed by the system to work properly (i.e. PMU minimal input power, supercapacitor loss current, etc.), the choice of the frequency (by the free space losses of the RF waves) and the propagation medium (now the air, next the reinforced concrete). The transmitted power is limited by regulations to 2 W EIRP (+33 dBm) for the ISM 868 MHz frequency band and the experiments were performed with lower powers. LoRa communications were impossible from the anechoic chamber because of the presence of RF absorbents but equivalence criteria to validate it are proposed.

               All this information was explicated all along the paper.

In this study, the main contribution and application of experiment results in concrete should be expressed clearly.

è          We currently have no conclusion about a real use in reinforced concrete because we received the reinforced concrete beam designed for experiments at the paper deadline. Moreover, we need an integrated and more compact prototype to perform efficiently experiments and it is in the design phase. The experiments in reinforced concrete which are very time-consuming will begin as soon as possible. As said, we decide to begin with in-the-air experiments and to introduce our work as a generic solution for CPS in SHM applications. In this paper, we also present the final objectives of the McBIM project and the first accomplished steps to meet them. Our first aim was to have a working prototype in-the-air to adapt and optimize it in more constraining environment.

               This information was explicated in the introduction and all along the paper.

This study mentioned the short-term observations for transmission, therefore, It should get strongly convincing and evidence in long-term observation.

è          Sorry, we do not understand this comment.

In this study, please describe in view of the relevance references to support it originality.

è          The relevant references to support our work have been more developed when referenced to highlight the originality and to have comparison elements. It is especially the case, when projects dedicated to the SHM of concrete are introduced (lines 95-114), when the comparative study of LoRaWAN autonomous sensing nodes is carried out (lines 162-173), when the choice of powering system is explained (lines 215-220) when the comparative study of wireless communication technologies is carried out (lines 363-374) and when the comparative study of the sensing nodes powered by RF system is carried out (lines 683-692).

               Moreover, some other changes and refinements were provided in order to answer the other reviewers’ comments, to precise some poorly defined points and to increase the English. In particular, the abstract, the introduction and the conclusion were redesigned as well as some figures were updated (Figures 3, 7, 10, 11 and 13) and one added (Fig. 5) and the references have been corrected ([7] and [11]) and some added ([3], [13], [14], [15] and [16]).

Reviewer 2 Report

Thank you for submitting your research work for review. I do have some comments to your submission. Please see below.

The authors are presenting a LoRaWAN battery-free wireless sensors network designed for structural health monitoring.  In summary, this network consists of a mesh grid of sensing nodes that collect physical data and communicating nodes that interface the sensing nodes with the digital world through the Internet. The sensing nodes can perform simultaneous temperature and relative humidity measurements and can transmit the measured data wirelessly over long-range distances by using LoRa technology.

The results of the paper revealed that the periodicity of the measurements and data transmission of the sensing nodes can be controlled by the dedicated RF source, by tuning the radiated power density of the RF waves.

Overall, the manuscript is well written. Structural integrity between the paragraphs and sections are observed to be strong.

However, there are some minor mistakes in the manuscript.

First of all, I think the label of the curve (blue or red) should be in the Figures. Besides, in the Figure 6a, I think the unit of DC voltage is wrong.

There has no unit of the last item in the Table 5 (Energy storage capacitance).

Authors should define the testing environment they use. Furthermore, they should motivate and characterize the testing activities they have carried out, as well as they should define what the objectives of this activity are.

As minor remark, a thorough proofreading is recommended, since the paper is affected by some typos and formatting issues.

Author Response

Dear Reviewer,

The authors would like to thank you for your given time and your helpful comments on this manuscript. We hope to have carefully addressed all the comments and revised the manuscript accordingly. The point by point replies to the comments and the main changes in the revised paper are summarized below.

Sincerely,

The Authors.

==========================================

Reviewer 2: Comments and Suggestions for Authors

Thank you for submitting your research work for review. I do have some comments to your submission.

Please see below.

The authors are presenting a LoRaWAN battery-free wireless sensors network designed for structural health monitoring.  In summary, this network consists of a mesh grid of sensing nodes that collect physical data and communicating nodes that interface the sensing nodes with the digital world through the Internet. The sensing nodes can perform simultaneous temperature and relative humidity measurements and can transmit the measured data wirelessly over long-range distances by using LoRa technology.

The results of the paper revealed that the periodicity of the measurements and data transmission of the sensing nodes can be controlled by the dedicated RF source, by tuning the radiated power density of the RF waves.

Overall, the manuscript is well written. Structural integrity between the paragraphs and sections are observed to be strong.

However, there are some minor mistakes in the manuscript.

First of all, I think the label of the curve (blue or red) should be in the Figures.

è          The figures 6, 11 and 13 have been updated by adding the label of the curve in the figures.

Besides, in the Figure 6a, I think the unit of DC voltage is wrong.

 è         Fig. 6 has been updated by changing mW by mV.

There has no unit of the last item in the Table 5 (Energy storage capacitance).

 è         Table 1 and Table 5 have been corrected by adding the unit (mJ) in the ‘energy storage capacitance’ field of the last item.

Authors should define the testing environment they use. Furthermore, they should motivate and characterize the testing activities they have carried out, as well as they should define what the objectives of this activity are.

è          To date, because it is the beginning of the project, the study is carried on for generic in-the-air applications (and not in reinforced concrete). The main objective is to provide a proof-of-concept of a wirelessly powered and battery-free wireless sensing node designed for cyber-physical systems dedicated to structural health monitoring applications -if possible in construction domain-. This proof-of-concept must be powered wirelessly over several meters, must sense some relevant environmental parameters (in a first time, temperature and relative humidity, but next other type of sensors could be integrated according to the targeted application) and must exchange data wirelessly (in a first time, for long range but other communication technologies are investigated according to the targeted application).

               Our final objective is to embed prototypes into reinforced concrete, but for ease of development, we decide to begin with in-the-air experiments and to introduce our work as a generic solution for CPS in SHM applications. Next step is to experiment in the targeted environment that says in a reinforced concrete beam, which will be more constraining for RF propagation. To date (March 2019) we have no conclusion about a real use in reinforced concrete but works/research (very time consuming) are under run.

               We currently have no conclusion about a real use in reinforced concrete because we received the reinforced concrete beam designed for experiments at the paper deadline. Moreover, we need an integrated and more compact prototype to perform efficiently experiments and it is in design phase. The experiments in reinforced concrete which are very time consuming will begin as soon as possible. As said, we decide to begin with in-the-air experiments and to introduce our work as a generic solution for CPS in SHM applications. In this paper, we also present the final objectives of the McBIM project and the first accomplished steps to meet them. Our first aim was to have a working prototype in-the-air to adapt and optimize it in more constraining environment.

               This information was explicated in the introduction and all along the paper.

As minor remark, a thorough proofreading is recommended, since the paper is affected by some typos and formatting issues.

 è         Several proof-readings were processed to correct them.

               Moreover, some other changes and refinements were provided in order to answer the other reviewers’ comments, to precise some poorly defined points and to increase the English. In particular, the abstract, the introduction and the conclusion were redesigned as well as some figures were updated (Figures 3, 7, 10, 11 and 13) and one added (Fig. 5) and the references have been corrected ([7] and [11]) and some added ([3], [13], [14], [15] and [16]).

Reviewer 3 Report

The paper presents a proof of concept of battery-free WSN for structural health monitoring based on LoRaWAN. The paper is quite well written, and the methodology is described with enough details. Some results obtained on a limited dimension network are presented and discussed to assess the reliability and performance of the proposed approach.

However, I have some concerns about the paper that have to be addressed before publishing the paper. First, I suggest I deep check of the paper to correct several typos (e.g. line 40: epmolyed, line 51: wall).and line 298: stat-up, line 416: the tfrom and so on) and, sometimes, English usage (e.g. redundancy such as line 104: two SN prototypes of sensing nodes). Acronyms have always to be defined (e.g. HFFS at line 165)

The Introduction Section should be enriched with some solutions including force and/or strain sensors already available in the literature. Moreover, pros and cons of literature solutions should be discussed with enough details; I suggest avoiding sentences as “Finally, the works 74 [7], [9], [10], and [11] provide incomplete solutions for some targeted applications by the project”. Why solutions are incomplete? Which part of your proposal overcome those limitations?

A suitable block diagram or electric scheme should be advisable to help reader in description of impedance matching circuit.

Y-axis Label in the first plot of Fig.6 has to be corrected (currently is DC Voltage [mW])

In Subsection 2.5, lines 247-255 the time events descriptions appear to be different to what shown in Fig.8. I guess it is only a writing problem, since a similar (repeated??) version of the same sentence is reported at the beginning of Subsection 2.7 and it is clearer and more flowing.

When describing the Adafruit DHT22 sensor, at line 287 the authors talk about its “step”. I suggest using the appropriate terminology; provided values are the resolution or the sensitivity (e.g. 0.1°C / LSB) of the sensor? Moreover, I guess that the authors should provide the required energy rather than the power for start-up and measuring; this way, is should be easier to understand the impact on the energy budget of 245 mJ.

The authors state in Subsection 2.7 lines 312-317 A photograph of the two SN prototypes manufactured are proposed in Fig. 9. At this stage of the research the SN prototypes are neither integrated or miniaturized but allow a proof-of-concept validation of the architecture highlighted in Fig. 4. Moreover, the choice in terms of sensors and communication technology do not provide a real low power solution. Nevertheless, having a functional complete SN prototype with the available sensor (not low power) and using LoRa technology and LoRaWAN protocol means that a more optimal system can be developed in the future”. However, when structural health monitoring is taken into account, load cells or attenuation from reinforced concrete are not secondary issues. It should be advisable for the authors to test their solutions in more realistic operating conditions; in my opinion, this is the main limit of the paper.

Picture in Fig.9 appears blurry and low-resolution. Authors are invited to improve it.

In Section 3 lines 336-347 the authors claim the equivalence of tests in anechoic chamber and “convenient location”; the reasons are not clear.

Readability of Fig.10 will enhance if the Y-axis of transmission periodicity is expressed in logarithmic scale.

In Eq. (2) and (3) the signs of multiplication have been accidentally replaced with the punctuation point.

Finally, the authors refer to papers [21] and [28] currently under review, making it difficult to understand the differences with the considered papers.

Author Response

Dear Reviewer,

The authors would like to thank you for your given time and your helpful comments on this manuscript. We hope to have carefully addressed all the comments and revised the manuscript accordingly. The point by point replies to the comments and the main changes in the revised paper are summarized below.

Sincerely,

The Authors.

Reviewer 3: Comments and Suggestions for Authors

==========================================

The paper presents a proof of concept of battery-free WSN for structural health monitoring based on LoRaWAN. The paper is quite well written, and the methodology is described with enough details. Some results obtained on a limited dimension network are presented and discussed to assess the reliability and performance of the proposed approach.

However, I have some concerns about the paper that have to be addressed before publishing the paper.

First, I suggest I deep check of the paper to correct several typos (e.g. line 40: epmolyed, line 51: wall).and line 298: stat-up, line 416: the tfrom and so on) and, sometimes, English usage (e.g. redundancy such as line 104: two SN prototypes of sensing nodes). Acronyms have always to be defined (e.g. HFFS at line 165)

 è         Several proof-readings were processed to correct them.

The Introduction Section should be enriched with some solutions including force and/or strain sensors already available in the literature. Moreover, pros and cons of literature solutions should be discussed with enough details; I suggest avoiding sentences as “Finally, the works 74 [7], [9], [10], and [11] provide incomplete solutions for some targeted applications by the project”. Why solutions are incomplete? Which part of your proposal overcome those limitations?

 è         The relevant references to support our work have been more developed when referenced to highlight the originality and to have comparison elements. It is especially the case, when projects dedicated to the SHM of concrete are introduced (lines 95-114), when the comparative study of LoRaWAN autonomous sensing nodes is carried out (lines 162-173), when the choice of powering system is explained (lines 215-220) when the comparative study of wireless communication technologies is carried out (lines 363-374) and when the comparative study of the sensing nodes powered by RF system is carried out (lines 683-692).

A suitable block diagram or electric scheme should be advisable to help reader in description of impedance matching circuit.

 è         A block diagram and a schematic of the entire composition of the rectenna have been added in Fig. 5.

Y-axis Label in the first plot of Fig.6 has to be corrected (currently is DC Voltage [mW])

 è         Fig. 6 has been updated by changing mW by mV.

In Subsection 2.5, lines 247-255 the time events descriptions appear to be different to what shown in Fig.8. I guess it is only a writing problem, since a similar (repeated??) version of the same sentence is reported at the beginning of Subsection 2.7 and it is clearer and more flowing.

 è         These sentences have been reformulated. The temporal quantifications of the events were removed from Subsection 2.5 to be only presented in Subsection 2.7.

When describing the Adafruit DHT22 sensor, at line 287 the authors talk about its “step”. I suggest using the appropriate terminology; provided values are the resolution or the sensitivity (e.g. 0.1°C / LSB) of the sensor? Moreover, I guess that the authors should provide the required energy rather than the power for start-up and measuring; this way, is should be easier to understand the impact on the energy budget of 245 mJ.

 è         The imprecise ‘Step’ word has been corrected by ‘resolution’ (line 380).

               In addition of the required power required by the sensor, its equivalence in terms of energy has been added (lines 381-384).

The authors state in Subsection 2.7 lines 312-317 “A photograph of the two SN prototypes manufactured are proposed in Fig. 9. At this stage of the research the SN prototypes are neither integrated or miniaturized but allow a proof-of-concept validation of the architecture highlighted in Fig. 4. Moreover, the choice in terms of sensors and communication technology do not provide a real low power solution. Nevertheless, having a functional complete SN prototype with the available sensor (not low power) and using LoRa technology and LoRaWAN protocol means that a more optimal system can be developed in the future”. However, when structural health monitoring is taken into account, load cells or attenuation from reinforced concrete are not secondary issues. It should be advisable for the authors to test their solutions in more realistic operating conditions; in my opinion, this is the main limit of the paper.

 è         The mechanical stress in reinforced concrete elements is another kind of parameter we would like to monitor by implementing a dedicated sensor (e.g. load cell) but we do not think it is the more priority task. A short discussion about the choice of the kind of sensor is carried out lines 697-701.

               To date, because it is the beginning of the project, the study is carried on for generic in-the-air applications (and not in reinforced concrete). The main objective is to provide a proof-of-concept of a wirelessly powered and battery-free wireless sensing node designed for cyber-physical systems dedicated to structural health monitoring applications -if possible in construction domain-. This proof-of-concept must be powered wirelessly over several meters, must sense some relevant environmental parameters (in a first time, temperature and relative humidity, but next other type of sensors could be integrated according to the targeted application) and must exchange data wirelessly (in a first time, for long range but other communication technologies are investigated according to the targeted application).

               Our final objective is to embed prototypes into reinforced concrete, but for ease of development, we decide to begin with in-the-air experiments and to introduce our work as a generic solution for CPS in SHM applications. Next step is to experiment in the targeted environment that says in a reinforced concrete beam, which will be more constraining for RF propagation. To date (March 2019) we have no conclusion about a real use in reinforced concrete but works/research (very time consuming) are under run.

               We currently have no conclusion about a real use in reinforced concrete because we received the reinforced concrete beam designed for experiments at the paper deadline. Moreover, we need an integrated and more compact prototype to perform efficiently experiments and it is in design phase. The experiments in reinforced concrete which are very time consuming will begin as soon as possible. As said, we decide to begin with in-the-air experiments and to introduce our work as a generic solution for CPS in SHM applications. In this paper, we also present the final objectives of the McBIM project and the first accomplished steps to meet them. Our first aim was to have a working prototype in-the-air to adapt and optimize it in more constraining environment.

               This information was explicated in the introduction and all along the paper.

               Finally, we agree, reinforced concrete behavior is -a priori- the main constraint of the project of providing communicating reinforced concrete based on RF transfer (power and data).

Picture in Fig.9 appears blurry and low-resolution. Authors are invited to improve it.

 è         Fig. 9 has been updated with higher resolution picture.

In Section 3 lines 336-347 the authors claim the equivalence of tests in anechoic chamber and “convenient location”; the reasons are not clear.

 è         Some precisions and reformulations have been operated to clarify that point. Because of the absorbents in the anechoic chamber, the LoRa communication can be achieved with the gateway which is not located in the anechoic chamber. Thus, to validate the functioning of the prototypes in the anechoic chamber, we must use the power consumption of the prototype, the duration of the transceiver use and the transmitted spectrum observed in the anechoic chamber during experiments. By comparing these parameters with the same obtained in outdoor experiments for which the LoRa communication is always achieved we estimate that we can conclude on the data transmission.

Readability of Fig.10 will enhance if the Y-axis of transmission periodicity is expressed in logarithmic scale.

 è         A log scale has been used in Fig. 10 and plot sizes have been increased.

In Eq. (2) and (3) the signs of multiplication have been accidentally replaced with the punctuation point.

 è         We correct that. Thank you.

Finally, the authors refer to papers [21] and [28] currently under review, making it difficult to understand the differences with the considered papers.

 è         The first referred paper initially under review [26] (initially [21]) is now accepted and published. It presents in details the implementation of the prototype of sensing node for SHM applications in harsh environments, but does not take into consideration -between others- the application targeted by the McBIM project, the comparison of 2D and 3D rectenna topologies, the experimentations in a simplified network topology and some discussions, especially regarding the temperature sensor choice. Thus the current paper completes the first by adding -between others- several applications complements, news experimental results and discussions.

               The second referred paper initially under review [33] (initially [28]) is now accepted but not yet published. It presents in details the characterization of the optimized rectenna with the reflector plan and does not treat the global system and the targeted applications.

               Moreover, some other changes and refinements were provided in order to answer the other reviewers’ comments, to precise some poorly defined points and to increase the English. In particular, the abstract, the introduction and the conclusion were redesigned as well as some figures were updated (Figures 3, 7, 10, 11 and 13) and one added (Fig. 5) and the references have been corrected ([7] and [11]) and some added ([3], [13], [14], [15] and [16]).

Round 2

Reviewer 3 Report

Revised version of the paper fully complies reviewer's concerns and suggestions.